# Self-Generating Data for Goal-Conditioned Compositional Problems

Ying Yuan[1], Yunfei Li[1], Yi Wu[1,2]

[1]Institute for Interdisciplinary Information Sciences, Tsinghua University; [2]Shanghai Qi Zhi Institute

## Abstract

*Building reinforcement learning agents that are generalizable to compositional problems has long been a research challenge. Recent success relies on a pre-existing dataset of rich behaviors. We present a novel paradigm to learn policies generalizable to compositional tasks with self-generated data. After learning primitive skills, the agent runs task expansion that actively expands out more complex tasks by composing learned policies and also naturally generates a dataset of demonstrations for self-distillation. In a proof-of-concept block-stacking environment, our agent discovers a large number of complex tasks after multiple rounds of data generation and distillation, and achieves an appealing zero-shot generalization success rate when building human-designed shapes.*

## 1. Introduction

Human everyday life involves many compositional decision-making problems that are composed of a sequence of subtasks. Developing intelligent agents with human-like capabilities of generalizing to a large number of compositional problems still poses challenges for reinforcement learning (RL).

Recent advances in reinforcement learning from offline datasets have demonstrated some generalization ability of the learned agent to unseen and long-horizon tasks [14, 2, 6]. However, the generalization relies on a pre-collected dataset with good coverage of rich behaviors [4, 10], which may not be easily scaled to compositional problems with a combinatorially large space. Therefore, it is natural to ask *can we develop an agent that self-generates datasets to teach itself how to solve compositional tasks?*

We introduce a novel way to solve compositional problems by iteratively generating data for self-teaching. In particular, we focus on the goal-conditioned RL setting [11]. Initially, we only provide the agent with demonstrations for learning primitive skills, e.g., how to interact with a single object. The agent then self-generates a dataset of trajectories demonstrating more complex tasks by composing previously learned skills and distills such compositional data

into its policy. After multiple rounds of data generation and distillation, the agent gradually masters a broad range of compositional tasks and can achieve non-trivial zero-shot generalization performance on complex problems.

To progressively generate compositional datasets, we propose a technique called *task expansion* that expands out more complex trajectories from already learned policies. Given a successful trajectory with initial state $s_0$ and terminal state $s_T$, task expansion generates new data by executing the current policy from $s_T$ to a new goal $g$ and appending the rollouts to the original trajectory. The new goal $g$ is selected to be out of reach from $s_0$ while easy to reach from $s_T$ under the current policy. Therefore, the self-generated dataset contains demonstrations for solving currently unreachable tasks with a higher level of compositionality (from $s_0$ to $g$). Our agent can then distill the strategies from the generated dataset to its policy to augment its ability for tackling compositional problems.

We experiment on a multi-object block-stacking domain. Starting from a dataset of trajectories containing single-object interaction only, our method can achieve a non-trivial success rate on zero-shot evaluation tasks after several rounds of expansion.

## 2. Related Work

**Offline reinforcement learning** Offline RL studies the topic of training an RL agent from a fixed dataset [13, 12, 9, 3]. Learning RL agents from offline datasets that could generalize to unseen and potentially long-horizon compositional tasks has attracted much research interest [14, 2, 10]. A notable line of work learns goal-conditioned agents from a dataset containing rich and diverse behaviors and could generalize to long-horizon tasks by stitching policies across different episodes [2]. Some recent works explicitly train a planner from offline datasets that can compose goal-conditioned policies to better deal with temporally extended problems [14, 5, 6]. Our method similarly starts from a behavior dataset, but instead of learning from the initial dataset only, it procedurally creates tasks and the corresponding datasets with a higher level of compositionally for self-teaching throughout training, which is an under-explored direction in offline RL.

**Curriculum generation** A bunch of literature in curriculum learning [1] (CL) for RL studies how to create a curriculum of subgoals/initial states to accelerate the convergence to the most challenging tasks [8, 7, 16, 15]. These methods propose to sample tasks with moderate difficulty for the current agent and result in a curriculum of tasks from easy to hard. Task expansion is technically similar to goal-generation methods, but with very distinct settings. CL typically assumes the prior knowledge about the most challenging tasks to solve, while our agent does not know the existence of any targeted compositional tasks a priori. Most goal-generation methods assume continuity of the difficulty level in the task space, meaning that the neighboring tasks share similar difficulty levels. However, the assumption may not hold true in strong compositional problems, where a slight difference in the task could result in significant changes in the task difficulty.

## 3. Preliminary

We implement our task expansion technique on a sparse-reward long-horizon environment called the multi-object block-stacking domain, where there exist $n$ cuboid blocks randomly placed on a desk initially and the task is accomplished only when all the $m$ goal objects are rearranged to their corresponding target positions stably. Note that the goal positions can be in the air, which requires some non-target blocks stacked below the goals.

We consider the setting of goal-conditioned Markov decision process with 0/1 sparse rewards, which is $(\mathcal{S}, \mathcal{A}, P(s'|s,a), \mathcal{G}, \mathcal{T}, r(s,a,g), \gamma)$. $\mathcal{S}$ is the state space, including the poses and the velocities of all the objects. $\mathcal{A}$ is the action space. In this work, each action selects one object and its desired pose and directly teleports that object. $\mathcal{G}$ is the goal space indicating the desired positions and IDs of a varying number of objects. $P(s'|s,a)$ indicates the probability of the transition from state $s$ to state $s'$ after taking action $a$, and $\gamma$ is the discounted factor. The reward function $r(s,a,g)$ is 1 only if the goal $g$ is reached stably at the current state $s$ within some precision threshold and otherwise 0. $\mathcal{T}$ is the task space represented as a set of paired initial states $s_0$ and goals $g$ from which to reset each episode. $\mathcal{T}$ varies among different rounds of task expansion. An episode terminates when either the goal is achieved or it reaches a maximum number of steps.

We adopt an actor-critic algorithm as the backbone RL agent, which trains a goal-conditioned policy $\pi_\theta(a|s,g)$ parametrized by $\theta$ and a universal value function [17] $V_\varphi(s,g)$. The objective of the RL agent is to find an optimal $\theta^\star$ that maximizes expected accumulated reward over the current task space, $\theta^\star = \arg\max_\theta \mathbb{E}_\mathcal{T}[\sum_t \gamma^t r(s_t, a_t, g)], a_t \sim \pi_\theta(\cdot|s_t, g), (s_0, g) \sim \mathcal{T}$.

## 4. Method

Our solution consists of three parts, including base policy learning, task expansion to generate compositional datasets, and policy distillation from the dataset. Once the base policy is learned, we alternatively operate task expansion and self-distillation so that our agent gradually teaches itself to handle more challenging tasks.

### 4.1. Learning the base policy

We aim to teach the agent primitive skills that could be further composed to solve complex problems in this phase. In the object manipulation domain we consider in this work, a natural choice is learning to interact with every single object. To this end, we collect a dataset of trajectories that only require transporting one object to reach success. The trajectories are actually single-step transitions since our action space directly works on the object level. We then train the policy using behavior cloning and finetune the policy with PPO [18] over all the tasks contained in the dataset.

### 4.2. Task expansion for data generation

In order to generate datasets for learning more complex tasks than the initial ones specified in the base policy learning phase, we propose the task expansion method, which consists of prospective task sampling, value-based task selection, and data augmentation.

In the $i$-th round of task expansion, we collect successful data trajectories using the policy checkpoint and the task buffer of the last round (the base policy and the initial tasks if $i = 1$). For each successful data trajectory, we randomly sample a set of $K$ new tasks under the uniform distribution of the number of goals, the positions of the goals, and the object indices corresponding to the goals, with the restriction that the newly sampled tasks should have only one alteration compared to the original task, i.e., either adding one goal or modifying one existing goal. We also experiment with a restriction-free version as an ablation, and we expect that the restriction helps the agent explore novel tasks more stably and well-organized.

To select from prospective tasks, given the initial state $s_0$ and the final state $s_T$ of each successful data trajectory, our main idea is to find a task $g^*$ that is relatively easy for the agent to accomplish starting from $s_T$ but relatively hard starting from $s_0$. In other words, both subtasks $s_0 \rightarrow s_T$ and $s_T \rightarrow g^*$ should be solvable by the current policy, while the task $s_i \rightarrow g^*$ is not trivial. Our implementation selects the best new goal $g^*$ based on a metric defined over the universal value function $V_\varphi(s,g)$ by

$$g^* = \arg\max_g V_\varphi(s_T, g) - V_\varphi(s_0, g). \quad (1)$$

We also experiment with other value-based selection metrics as an ablation.

After selecting the new goal $g^*$, we verify whether it can be reached from $s_T$ by executing the current policy in the environment. If the execution is successful, we can get a demonstration for the task $s_0 \rightarrow g^*$. Otherwise, we relabel the whole trajectory as targeting the state where the failed execution ends up. Such relabeling boosts data efficiency and guarantees that all the generated tasks can be accomplished starting from the original final state in the collected data trajectories.

### 4.3. Learning generated tasks with self-distillation

After collecting a dataset $\mathcal{D}$ of trajectories to solve newly generated tasks, we run behavior cloning (BC) to distill into the policy. Since previous successful trajectories are repurposed as good prefixes to solve more complex problems in $\mathcal{D}$, BC on this dataset effectively teaches the agent to break down hard tasks using previous policies. In our environment, we empirically find that newly proposed tasks tend to have higher goals on top of the existing objects, so the data trajectories hint at setting up the base objects to accomplish the high goals in the air. We again adopt PPO[18] to finetune our policy on generated tasks so as to revise minor errors after BC due to the fact that some trajectories may include detours and are not exactly the prefix of novel tasks.

## 5. Experiment

In this section, we report the preliminary results of our self-generation paradigm in a block-stacking domain. We present how our method gradually generates more compositional data with multiple rounds of expansion and achieve non-trivial zero-shot generalization performance.

In our implementation, we set the maximum number of goals as the total number of cuboid blocks $n$. In order to familiarize the agent with a broader variety of initial tasks, we also create some multiple-goal tasks based on the original one-goal tasks by assigning additional goals to be the object indices and current positions of certain non-goal objects, which we expect the agent can handle using the base policy checkpoint. Note that the number of goals $m$ in such tasks is set to be no larger than 3, so we can test the generalization ability of our method and verify that the agent can explore and learn novel tasks with more goals to achieve that have not been seen during training.

### 5.1. Statistics of task expansion

We summarize the statistics of generated tasks in each round of the task expansion process. Figure 1 shows the distribution of the maximum height of the multiple goals in each task for each round of task expansion respectively, which presents a tendency towards higher goals as the number of rounds increases. Figure 2 shows the distribution of the number of goals in generated tasks for each round,

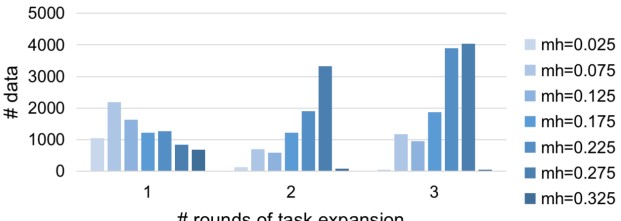

Figure 1. Distribution of maximum heights (mh) of generated tasks in different rounds of task expansion.

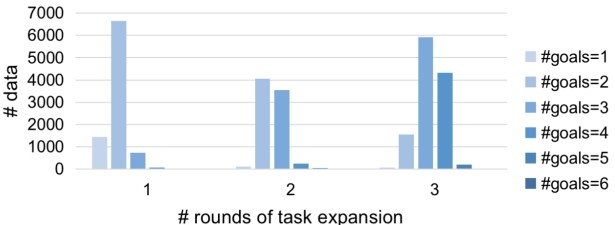

Figure 2. Distribution of the number of goals (#goals) of generated tasks in different rounds of task expansion.

| # rounds | # high goals | ep_len_mean |
|----------|--------------|-------------|
| 1 | 12 | 2.5 |
| 2 | 18 | 5 |
| 3 | 938 | 7.6 |

Table 1. Statistics of generated tasks in different rounds of task expansion. "# high goals" stands for the number of tasks with at least 3 goals above ground. "Ep_len_mean" stands for the mean episode length till success or termination.

| # rounds | "I-shape" sr. | "Y-shape" sr. | "3T-shape" sr. |
|----------|---------------|---------------|-----------------|
| 1 | 93.7% | 1.5% | 58.2% |
| 2 | **95.6**% | 33.5% | 93.5% |
| 3 | 89.3% | **43.8**% | **98.2**% |

Table 2. Success rate of zero-shot evaluation on manually designed tasks using policies trained in different rounds of task expansion.

which indicates the number of goals gets larger as our algorithm proceeds. Note that our initial task space only contains tasks with no more than 3 goals, while the task expansion process finds a large proportion of tasks with 3 or 4 goals, and even some with 5 goals. Our agent manage to achieve a 99% success rate on the generated tasks in each round within 150M timesteps of PPO tuning. As is shown in Table 1, the number of "high goals" proliferates in the third expansion and the mean episode length scales up gradually, showing that the generated tasks are not trivial and become more complicated in each round of exploration.

### 5.2. Zero-shot generalization to held-out tasks

After several rounds of task expansion and self-teaching, we test whether our agent can generalize to compositional tasks that are never provided to it before. Specifically, we

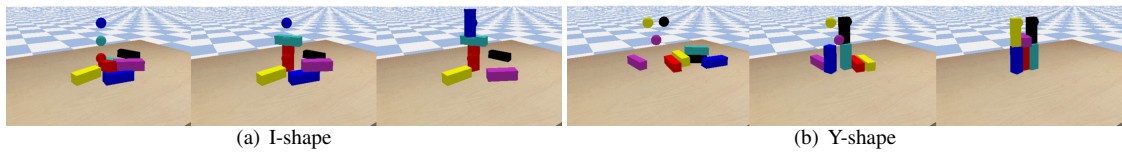

(a) I-shape  (b) Y-shape

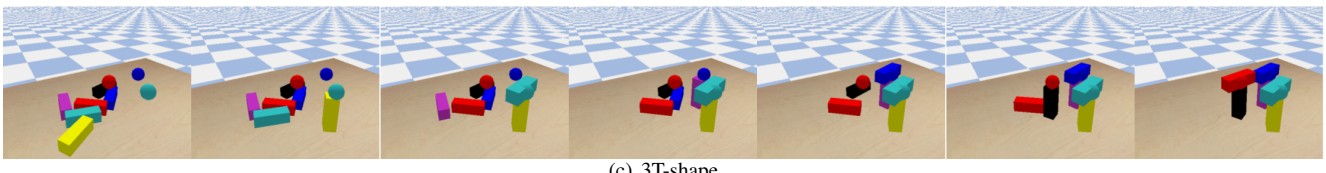

(c) 3T-shape

Figure 3. Visualization of three categories of held-out evaluation tasks.

design three categories of evaluation goals, the "I-shape", the "Y-shape" and the "3T-shape", as shown in Figure 3, and test the agent from initial states with all blocks randomly scattered on the desk. The policy checkpoints tuned for 150M timesteps after each round of task expansion are selected for evaluation. Each checkpoint is tested for 4096 environment steps for each category of goals. As is reported in Table 2, the zero-shot performance on the three categories of held-out tasks gets better in later rounds. Empirically, our agent is not an expert at stacking high goals, which needs to build bases first, but later it can handle such tasks. The strategy of our agent is illustrated in Figure 3.

### 5.3. Ablation studies and comparisons

To evaluate how much our algorithm outperforms other implementations, we do ablation experiments to answer these questions: 1) Does our task expansion solution outperform directly running RL on the evaluation tasks with the base policy? 2) Does the restriction of sampling new goals with only one alteration from the original tasks necessary for better zero-shot performance? 3) What if we use alternative metrics to select tasks from all the sampled ones?

We try a naive approach that directly trains over the evaluation tasks with PPO after learning the base policy. The success rate is always zero in 137M timesteps. Therefore, task expansion can be viewed as an automatic curriculum that smooths out the challenges of directly optimizing strong compositional problems under sparse reward.

We experiment with removing the restriction of only allowing one goal to be different from the original task, that is, sampling new goals randomly in the goal space ("w/o restr."). As shown in Table 3, it performs worse than our presented method, possibly because goal sampling with the restriction could generate more feasible tasks.

We study other metrics to select the best new goal in task expansion by choosing $\arg\max_g V_\varphi(s_T, g)$ ("end2new")

| Methods | "I-shape" sr. | "Y-shape" sr. | "3T-shape" sr. |
|---|---|---|---|
| w/o restr. | 0.3% | 4.1% | 38.7% |
| init2new | 7.3% | 0.0% | 0.0% |
| end2new | 79.4% | 20.3% | 59.4% |
| ours | 93.7% | 1.0% | 58.2% |

Table 3. Success rate of zero-shot evaluation compared with other design choices. All variants use the policies tuned for 150M timesteps after the *first* round of task expansion for evaluation.

or $\arg\min_g V_\varphi(s_0, g)$ ("init2new"). As is shown in Table 3, "init2new" generalizes significantly worse, while "end2new" achieves comparable performance with our metric. These results imply that generating new tasks that are feasible by composing previous policies is more critical to the success of the self-generation paradigm than simply creating challenging novel tasks.

## 6. Conclusion

We present a framework that allows an RL agent that automatically generates datasets using its already learned skills to teach itself how to solve compositional problems. We propose a powerful technique, task expansion, to enable generating more complex tasks and their solutions by composing previously learned tasks. With self-distillation from the generated solutions, our agent can gradually learn to solve problems with stronger compositionality, and achieves promising zero-shot evaluation performance in a sparse-reward block-stacking domain. We are still working to extend this framework to more realistic domains such as robotic manipulation, and we believe leveraging self-generated data is a promising direction for scaling up RL to the complexities of real-world problems.

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
