# OpenReview forum: "Self-Generating Data for Goal-Conditioned Compositional Problems"
_ICLR.cc/2023/Workshop/RRL — RRL 2023 Poster_

### Official Review · Reviewer_BQtC · 2023-02-22
**Interesting idea, but the presentation needs to be improved**

**Rating:** 2
**Confidence:** 3

**Review:**

This paper studied compositional problems where the task requires multiple ordered steps to complete. The authors proposed a novel algorithm that involves an iterative approach between using the learnt skills to generate new tasks and learning skills from those tasks. The process proposed in this paper forms an automatic curriculum for the agent to learn from. This paper aligns with the theme of the workshop in that it reuses learned skills or policies to generate new datasets during each iteration.

In general, I find the idea of this paper interesting. The proposed method demonstrates good performance on a simplified setting. However, the presentation of the paper needs to be considerably improved for ease of comprehension from the reader's perspective. More details and clarification needs to be made before being published.  Here are some suggestions that I think the authors can work on or make clarifications:

1. line 130: The task considered in this paper is not really long-horizon. From Table 1, the mean length of the trajectories is only 7.6.
2. line 140: From my understanding, the velocities of the objects are not needed since you are using teleports as actions.
3. I am very confused about how the goal is represented as inputs to the network since their size can vary during the course of learning.
4. How the $g^*$ being optimized from eq. (1) should be clarified. I think the choice of optimization algorithm here that deals with both discrete (the goal object) and continuous variables (the position of the goal) is important to the performance of the algorithm.
5. line 219: HER [1] should be cited here for the relabeling trick.
6. line 253: "by assigning .... certain non-goal objects". I do not fully understand what the author means from here. Does that mean you set object B's current position to be the target of object A so that the agent needs to move B somewhere else first, then they can move A to the target?
7. The success rate for the "Y-shape" in the first line of the Table 1 should be the same as in the fourth line of Table 3 from my understanding. Maybe there is a typo?
8. The differences between this work and the existing methods need to be discussed, for example [2].

[1] Andrychowicz et. al., Hindsight Experience Replay, NIPS 2017

[2] Li et. al., Solving Compositional Reinforcement Learning Problems via Task Reduction, ICLR 2021

---

### Official Review · Reviewer_vPHa · 2023-02-27
**A novel idea for a general problem but with a task-specific method. Well-written but with the potential to improve clarity and evaluation.**

**Rating:** 2
**Confidence:** 3

**Review:**

Summary:

- This paper focuses on compositional goal-conditioned MDPs. When we don't have high-quality data or prior information about the task of interest, the authors argue that current offline RL or curriculum learning methods fail to work. Instead, this paper proposes a novel approach to generate datasets demonstrating more complex behaviours in the environment based on the universal value function through iterative data expansion and policy-distillation steps.
The paper empirically evaluates the quality of learned policy in a multi-objective block-stacking task. It shows this method learns policies that can solve more complex tasks and generalize to unseen tasks.

Strength:
- The paper proposes a novel progressive compositional data generation method using the universal value function.
- It systematically evaluates the quality of the proposed method in a compositional goal-conditioned task and studies the contribution of different components of the proposed method in a set of ablation studies.
- The paper is well-written and easy to follow.

Weaknesses:
- The paper discusses a broad range of problems initially, but as early as section 3, it becomes too focused on a specific environment. This restricts the applicability of the main contributions to only that particular task. For instance, section 4.1 should discuss the proper definition of primitive skills for a general environment, while section 4.2 should address how to define the restriction on newly sampled tasks in such an environment. The sections before section 5 should be task-independent to increase the paper's contribution beyond the limited scope of the specific task.
- The ablation study conducted to verify if online RL algorithms can solve the problem should be more extensive. Firstly, starting from primitive skills before switching to the main task seems might induce a negative bias in the policy, did you try to train it on the evaluation task from scratch? Secondly, PPO was only trained with 137M timesteps while the proposed method is trained for much more time steps. The fine-tuning of PPO was done with a budget of 150M steps in each iteration and as vaguely mentioned in the paper the results were obtained after "several iterations". I would suggest reporting the performance of vanilla-PPO trained with the same number of environment transitions, at least as a baseline.
- In Equation 1, for goal g* to have the right difficulty level (solvable $s_T \rightarrow g*$ but not trivial $s_0 \rightarrow g*$), the universal value function $v_{\varphi}$ should be a close approximation of the true universal value function of current policy, and it needs to be approximated at every iteration of expansion-distillation. How is it learned efficiently? Do you have an approach to using collected data to learn it or is it based on online interaction with the environment?

Suggestions for improvement:
- In section 3, an example of what is T in the definition of MDP and how it changes through different rounds of task expansion can help to improve the clarity of the paper.
- Although the paper aims to address the lack of a dataset containing rich behaviours, it still appears to be an issue in this method. As described in line 250-forward, having multiple-goal initial tasks is important for the success of this method. It would be helpful to show the sensitivity of this method to the quality/diversity of initial tasks.
(on a related note, I would suggest adding the data of initial goals to Figure 2 as well.)
- Figure 3 lacks a clear explanation of each image in its respective section.